# Breastfeeding in China: A Review of Changes in the Past Decade

**DOI:** 10.3390/ijerph17218234

**Published:** 2020-11-07

**Authors:** Qing Li, Jianli Tian, Fenglian Xu, Colin Binns

**Affiliations:** 1Department of Nursing, Chengde Medical University, Chengde 067000 China; liqing8168@cdmc.edu.cn; 2Data Analysis & Surgical Outcomes Unit, Royal North Shore Hospital, Sydney, NSW 2065, Australia; Fenglian.Xu@health.nsw.gov.au; 3School of Public Health, Curtin University, Perth, WA 6845, Australia

**Keywords:** breastfeeding, prevalence, duration, review, China

## Abstract

This review summarizes breastfeeding rates in China reported during the decade 2007–2018, a decade on from our previous review published in 2007. Compared with the studies undertaken before 2007 in China, recent studies are more likely to report breastfeeding rates using longer periods of observation, enabling rates to be summarized to six and 12 months postpartum in this review. There appears to have been a modest increase in breastfeeding in China. The mean duration of “any breastfeeding” was 10 months (9 to 11 months in the majority of cities), an increase compared with the previous review in which the mean of “any breastfeeding” duration was 8 months (7 to 9 months in the majority of cities). Using data from cohort studies, the proportion of infants being breastfed at 4 months increased from 78% in the earlier decade to 83% more recently. A second baby is usually breastfed for longer than the first, considering both “any” and “exclusive breastfeeding”. China is a huge country and there is considerable diversity in culture, level of economic development, education and breastfeeding rates in different areas of China, but our review suggests that there has been some improvement in the “any breastfeeding” rate in the most recent decade.

## 1. Introduction

Breastfeeding (BF) provides for optimal nutrition and development parameters, including improved functioning of the immune system, for infants until about six months of age and then into adult life [1,2]. The WHO (World Health Organization) and UNICEF (The United Nations Children’s Fund) recommend that babies are breastfed exclusively to around six months and continue to be breastfed after the introduction of complementary food [3]. Increasing “exclusive breastfeeding” in the first six months of life to at least 50% across the globe is included in the WHO Nutrition goals for 2025 [4]. A new target set in the National Program of Action for Child Development in China from 2011 to 2020 is an “exclusive breastfeeding” rate of 50% and over at the sixth month of life. This is more specific than the previous target from 2001 to 2010, which simply stated a breastfeeding rate of 85%, without specifying exclusivity [5].

We published a literature review in 2009 that summarized breastfeeding rates, duration and reasons of discontinuation of breastfeeding in China, including information on an increase of breastfeeding rates as a result of initiatives introduced to promote breastfeeding beginning in the 1990s [6]. Few cities and provinces reached the national target of an “exclusive breastfeeding” rate of 80% at four months after birth by 2000 [6]. Since 2007, more breastfeeding initiatives and interventions have been taken to promote breastfeeding in China. Maternity leave has been increased from 90 days to 120–180 days in recent years. The “universal two-child policy” has also been implemented throughout the country since 2016. It is time to review the changes of breastfeeding in China for the last decade.

The aim of this review is to document breastfeeding rates in China in the decade since 2007, including the changes of breastfeeding rates, duration and reasons for discontinuing breastfeeding. This review will update our previous review.

## 2. Materials and Methods

A literature search was undertaken using the Chinese databases: China National Knowledge Infrastructure (CNKI) and Wan Fang Medical Data and the English databases: Medline, Science Direct and ProQuest from 2007 to 2017. The databases were searched using the following key words: China, breastfeeding, breast feeding, breast-feeding and infant feeding. As shown in Figure 1, a total of 2894 studies were obtained, and ultimately 91 studies (including 49 cohort studies and 42 other studies) were analysed for this review. We assess the methodological quality of the selected studies based on the criteria proposed in Strengthening the Reporting of Observational Studies in Epidemiology (STROBE) [7] and the requirements of this study. All studies included met the following criteria: (1) sample size greater than 200; (2) loss to follow-up or incomplete records less than 20%; (3) for prospective studies, more than three follow-up interviews with 1–2 month intervals; (4) breastfeeding rates calculated using correct definitions and statistical methods; and (5) published in peer-reviewed journals.

All of the studies were assessed for quality using a method based on the STROBE criteria and used by us in a previous meta-analysis in China [8]. This method allocates a maximum score of 18, and papers that met a minimum score of 11 were included.

### Definitions of Breastfeeding

WHO breastfeeding definitions are used in this paper [9].

“Exclusive breastfeeding”: Breastfeeding while giving no other food or liquid, not even water, with the exception of drops or syrups consisting of vitamins, mineral supplements or medicines.

“Full breastfeeding”: Exclusive breastfeeding or predominant breastfeeding or almost “exclusive breastfeeding”. Breastmilk is the only source of milk given to the infant regardless of supplementation with other fluids such as water and orange juice.

“Any breastfeeding”: The child has received breastmilk (direct from the breast or expressed) with or without other drink, formula or other infant food. “Any breastfeeding” included “exclusive breastfeeding” and “full breastfeeding” and “partial breastfeeding”.

In some papers, only the term “breastfeeding” was used. The “breastfeeding” is categorized as “any breastfeeding” in this review.

## 3. Results

### 3.1. Breastfeeding Rates in China from Cohort Studies

Table 1 summarizes “exclusive breastfeeding” and “any breastfeeding” at six months in seven large, nine medium-sized and four small cities in China. The cities included in Table 1 are Beijing [10], Shanghai [11], Chengdu [12,13], Yinchuan [14], Changsha [15], Xi’an [16], Guangzhou [17], Ka’shen [18], Jinzhou [19], Shenzhen [20], Ake’su [21], Luzhou [22], Jiangyou [23], Ma’anshan [24], Mianyang [25], Ningbo [26], Wenling [27], Lishui [28], Yongkang [29] and Longnan [30]. The results are from 21 papers in Chinese and English published from 2007 to 2017 (Table 1).

The cohort studies in Table 1 were followed-up to six months and longer in 17 cities and to four months and over in three cities. Breastfeeding rates at six months from cohort studies are shown in Figure 2.

Table 1 shows that the “any breastfeeding” rate at six months ranged from 43.13% to 95.50% in 17 cities (excluding Yingchuan, Ma’anshan and Ningbo where the follow-up time was less than six months). Of the 17 cities, the “any breastfeeding” rates at six months were above 85% in AKesu and Ka’shen of Xinjiang Uygur Autonomous Region; between 80% and 81% in Luzhou and Mianyang of Sichuan province, and Wengling of Zhejiang province; and below 80% in other 12 cities. In majority cities (15/17 = 88%), the “any breastfeeding” rates at sixth months were below 85%.

Table 1 also shows that “exclusive breastfeeding” rates at six months were between 0.50% and 33.45% in 17 cities (excluding Yingchuan, Ma’anshan and Ningbo where the follow-up times were less than six months). The “exclusive breastfeeding” rates were below the target of 50% “exclusive breastfeeding” rate at six months [5]. In Akesu, a medium-sized city from Xinjiang, 96% babies were exclusively breastfed before discharged from hospital for birth, but only 0.5% babies were still exclusively breastfed at six months after birth [21]. The low “exclusive breastfeeding” rate in Akesu was consistent with a report (based on a study in 2003–2004) from Shihezi, Xinjiang [31]. The average “exclusive breastfeeding” duration in Xinjiang was 1.8 months [32].

“Any breastfeeding” rates before discharge from hospitals were between 82.09% and 99.50% (reported from six studies in six cities in Table 1). “Exclusive breastfeeding” rates before discharge from hospitals showed a wide variation from 28.70% to 96.00% (reported from five studies in five cities in Table 1).

“Any breastfeeding” rates at the 12th month ranged from 10.97% to 73.26% [11,23,33,34,35] (from five studies in five cities in Table 2). Less than 1% (0.26–0.38%) of children were breastfed to two years [34,35].

In addition to the studies in Table 1, a cohort study (*n* = 1350; 2013–2014) in rural areas from three cities in Shaanxi Province, showed that the “any breastfeeding” rates were 67.85% and 39.41% at six and 18 months, respectively, and the “exclusive breastfeeding” rate at six months was 35.04% [33]. Another cohort study from poor areas in Anhui province in 2012–2013 showed that the “any breastfeeding” rates were 75.45%, 71.02%, 27.4% and 0.38% at four, six, 12 and 24 months, respectively [34]. A study in Guangzhou (2013–2014) showed that “any breastfeeding” rates were 65.49%, 47.58%, 10.97% and 0.26% at four, six, 12 and 24 months, respectively [35]. A cohort study in Zhoushan, a medium-sized city in Zhejiang Province, showed that “full breastfeeding” rates were 87.3%, 68.5%, 48.1%, 26.2% and 5.6% at one, three, six, nine and 12 months, respectively, in 2002–2015 [36]. A notable improvement in breastfeeding cohort studies in China was that follow-up time was extended to six months in more studies compared with those before 2007 [6].

The breastfeeding rates from three cities (Beijing, Guangzhou and Luzhou) were reported both in the previous literature review and the current review although the follow-up times were different. Compared with our previous literature review, “any breastfeeding” rates in Beijing increased significantly. For example, the “any breastfeeding” rate in Beijing was 95.00% (95% CI = 92.75–97.25) at one month and 89.72% (95% CI = 86.58–92.86) at four months in 2007–2009 [10] compared to 84.00% (95% CI = 76.8–91.2) and 76.00% (95% CI = 67.6–84.4) in 1997 [6]. “Exclusive breastfeeding” rates at one and four months in 2007–2009 in Beijing were not statistically different from those in 1997 [6,10]. “Any breastfeeding” rates at three months in Guangzhou also increased compared with our previous literature review from 93.2% (95% CI = 91.8–94.6) in 1998–1999 [6] to 97.12% (95% CI = 95.51–98.73) in 2005–2006 [17]. The “exclusive breastfeeding” rate at one month in Guangzhou decreased to 72.12% (95% CI = 67.81–76.43) in 2005–2006 [17] from 90.5% (95% CI = 88.9–92.1) in 1998–1999 [6]. “Any breastfeeding” rates at one month and “exclusive breastfeeding” rates at three months in Guangzhou in 2005–2006 were not statistically different compared with those in 1998–1999. On the other hand, in Luzhou, “any breastfeeding” rates at one month decreased [6,22]. The “any breastfeeding” rate at one month in Luzhou was 92.59% (95% CI = 90.26–94.92) in 2012 [22] and 97.5% (95% CI = 95.4–99.6) in 2002 [6]. The “exclusive breastfeeding” rates at one month and three months in Luzhou decreased significantly compared with previous literature review [6,22]. The “exclusive breastfeeding” rates in Luzhou were 45.06% (95% CI = 40.64–49.48) at one month and 42.8% (95% CI = 38.4–47.2) at three months in 2012 [22]; while the “exclusive breastfeeding” rates were 89.6% (95% CI = 85.4–93.8) at one month and 83.7% (95% CI = 78.6–88.8) at three months in 2002 [6].

Table 2 summarizes the “any breastfeeding” rate at 12 months from five cohort studies.

### 3.2. Breastfeeding Rates in China from Other Types of Studies

Table 3 summarizes “exclusive breastfeeding” and “any breastfeeding” rates at six months and breastfeeding initiation rates in ten large cities and four provinces in China. The cities and provinces included in Table 3 are Tianjin [37,38], Shanghai [39,40], Chongqing [41,42], Zhengzhou [43,44], Wuhan [45,46], Guangzhou [47,48], Changsha [49], Xining [50,51], Changchun [52,53], Nanchang [54], Zhejiang [55,56], Shanxi [57,58,59], Anhui [34,60,61] and Jiangsu [62,63]. For the provinces or cities where more than one study was carried out, the most recent results have been presented. The research methods included cross-sectional and retrospective studies and cohort studies that did not satisfy the criteria for Table 1.

Table 3 shows that breastfeeding initiation rates were between 77.02% and 98.00%; “any breastfeeding” rates at six months were between 63.41% and 92.93%; and “exclusive breastfeeding” rates at six months were between 17.87% and 58.50%. Of the 10 cities and four provinces in Table 3, the exclusive breastfeeding rate at six months was above 50% (the national goal) in three cities (Wuhan, Shanghai and Guangzhou) (3/14 = 21%) and the “any breastfeeding” rate at six months was above 85% (the previous national goal) in three cities and one province (Nanchang, Changchun, Chongqing and Shanxi) (4/14 = 29%). The lower number reaching the “exclusive breastfeeding” target needs improving.

Furthermore, the studies in Table 3, in a cross-sectional study (*n* = 1288) from 32 maternity and child healthcare hospitals in China, the rate of “any breastfeeding” rate at discharge (in hospital) was 96.4% and the “exclusive breastfeeding” rate at discharge was 46.6% in 2010 [64]. A major survey (*n* = 14,262) of rural infants from 5 provinces in Western China in 2009 found that the “any breastfeeding” rate at discharge was 97.46% and the “exclusive breastfeeding” rate at four months was 52.02% in these rural areas [65]. A cross-sectional survey in 12 provinces in central and western China in 2010 showed that the overall “any breastfeeding” rate was 98.3%, and the overall “exclusive breastfeeding” rate at six months was 28.7% [66]. A national representative survey (*n* = 14,458) from 55 counties in 30 provinces in China (2013) showed that the crude “exclusive breastfeeding” rate at six months of age was 20.73% (908/4381), and the weighted “exclusive breastfeeding” rate was 18.6% [67]. These studies showed that “exclusive breastfeeding” rates at six months were below 30% (20.73–28.70%).

A recent survey found that “exclusive breastfeeding” rates at six months in Beijing (*n* = 890) [68], Shanghai (*n* = 5672) [69] and Dalian (*n* = 32,466) [70] were 31.82%, 51.34% and 55.79%, respectively. In Shenzhen, “exclusive” and “any breastfeeding” rates at six months were 61.2% and 91.3%, respectively (*n* = 1000) in 2015 [71]. A survey (*n* = 1019) in nine community hospitals from three cities in Anhui province showed that the overall “exclusive breastfeeding” rate at six months were 61.9%, including 64.5% in Hefei, the capital city of Anhui, 60.7% in Fuyang and 59.5% in Wufu [72]. The “exclusive breastfeeding” rates reported in these recent studies (2016–2018) show improvement. This may be associated with the promotion of breastfeeding in recent years [68,69,70,71,72].

### 3.3. Reasons for the Variation in Reported Breastfeeding Rates

This review shows that breastfeeding rates from studies conducted in different cities and areas in China can show major differences. The large range in breastfeeding rates may be associated with differences in breastfeeding promotion strategies, study methods, study populations, culture and other factors including mother’s age, educational background, household income, residential area, delivery mode and family support. For example, in cross-sectional or retrospective studies, the methods used to assess infant feeding might be in the “last 24 h prior to the survey” or “during the last month”.

### 3.4. Length of Breastfeeding and Proportion Breastfeeding in China from 2007 to 2017

The mean duration of “any breastfeeding” between 2007 and 2017 was between eight and 12 months (9 to 11 months in the majority of cities) reported from cross sectional studies. [23,34,44,50,73,74,75,76,77,78,79,80]. The mean duration was 10.1 months (weighted by provincial population) from eight cross-sectional studies. In the previous study, the weighted mean duration of breastfeeding was 8.0 months from cross-sectional studies. The reported mean duration of breastfeeding shows an increase of almost two months compared with that in the previous review (7 to 9 months in majority cities) [6]. For example, “any breastfeeding” duration in Shanghai was 7.4 months in between 1999 and 2002 [6] and has increased to 8.45 and 9.98 months (Figure 3 and Figure 4). The mean durations of “any breastfeeding” reported from studies before and after 2007 are shown in Figure 5. This shows that the majority of studies reporting longer durations of breastfeeding have been published since 2007.

The proportion of mothers breastfeeding at four months was reported in both decades in the cohort studies. The weighted (on provincial population) mean rate of four months “any breastfeeding” was 78.2% before 2007 and 83.0% more recently.

A cross-sectional study (*n* = 635, 2017) undertaken in four hospitals in Shandong province reported the longest mean duration of “any breastfeeding” of 12.0 ± 0.82 months. The mean “exclusive breastfeeding” duration was 4.01 months [73]. In this study, all mothers participating were nurses [73]. A cohort study in 17 cities from Shandong province (*n* = 1630) showed that the average duration of “exclusive breastfeeding” was 6.15 ± 2.35 months (mean ± SD) in 2008–2013 [76]. Another cohort study at Lianyungang city, Jiangsu province (*n* = 383), indicated the average duration of “exclusive breastfeeding” was 5.98 ± 4.43 months (mean ± SD) in 2013–2014 [78]. The studies from other cities showed that the length of “exclusive breastfeeding” were between 2.9 and 5.36 months [44,75,77,80]. The study in Shihezi city (*n* = 200, 2007–2008), Xinjiang province, showed the shortest duration of “exclusive breastfeeding” (0.92 month) [79].

The reasons for the increase in breastfeeding duration may include policy changes; the increase in maternity leave [81,82], institution of a “universal two-child policy” [83,84] and more breastfeeding promotion, including using mobile telephones to send educational messages regularly to mothers [85,86]. Maternity leave with full salary was extended from 90 days to 98 days in 2012 in China [81] and from 98 days to 128–180 days in the majority provinces from 2016 when the “universal two-child policy” was implemented throughout the country [82]. Fathers have nursing leave (paternity leave) of 7 to 30 days with full salary since 2016 [82]. Employers provide mothers with one hour for breastfeeding during work time [81]. With the policy support, mothers and families have less financial burden and more time for breastfeeding [84].

The two-child policy has been gradually implemented in China since 2011 [83,84,87]. Couples have been allowed to have the second baby if both of them come from a single-child family since 2011, if at least one of them come from single-child family since 2013 and unconditionally since 2016 (the “universal two-child policy”) [83,84]. Chinese citizens from ethnic groups are not subject to these limits. Studies showed that the second babies have longer breastfeeding duration (for both “any” and “exclusive breastfeeding”) than the first [74,80]. A survey in Hangzhou and Shanghai in 235 multiparous mothers (2017) showed that both “exclusive breastfeeding” and “any breastfeeding” durations for the second baby were longer than for the first one [80]. “Exclusive breastfeeding” duration was 3.28 months for the first baby and 3.61 months for the second baby (*p* < 0.05); and “any breastfeeding” duration was 9.78 months for the first baby and 10.88 months for the second baby for (*p* < 0.01) [80]. Similarly, a retrospective study (*n* = 168) in Zhejiang province in 2017 found that “exclusive breastfeeding” duration and “any breastfeeding” duration of the second baby were longer than those of the first (*p* < 0. 05) [74]. “Any breastfeeding” duration was 9.8 ± 3.7 months for the first baby and 11.6 ± 4.6 months for the second [74]. “Exclusive breastfeeding” duration was 2.9 ± 2.4 months for the first baby and 3.3 ± 2.3 months for the second [74]. The reason for the increase in breastfeeding duration of the second baby may associated with the mother’s breastfeeding experience from the first baby, more involvement of fathers or more supportive breastfeeding policy [82].

In addition, breastfeeding education using the internet and mobile apps increased breastfeeding knowledge of mothers, fathers and their families [65,72,88]. A study in Shanghai found that receiving a weekly text message supporting breastfeeding significantly improved the “exclusive breastfeeding” rate (from 6.3% to 15.1%) and increased the duration of “exclusive breastfeeding” assessed at six months (from 8.87 to 11.41 weeks) [85]. The use of mobile applications appears to be an effective way to support breastfeeding and increase “exclusive breastfeeding” rates [86]. Participating in “pregnancy school” (antenatal classes) was also associated with increased breastfeeding duration [80]. Other risk factors associated with breastfeeding duration included mother’s education level (mothers with high education level were more likely to be employed and had shorter breastfeeding duration), grandparents’ support (babies were breastfed for a shorter duration if grandparents helped to take care of them) and having breastfeeding difficulties (babies were less likely to be breastfed if their mother had breastfeeding difficulties) [80].

### 3.5. Breastfeeding in Minority Areas or Groups

In the Xinjiang Uygur Autonomous Region, the “exclusive breastfeeding” rate and “any breastfeeding” rates at one and a half months were 57.65% and 93.04%, respectively, in the city of Yili in 2016 [89]. A cross-sectional study showed the “exclusive breastfeeding” rate at four months to be 63.03% in Tuoli between 2013 and 2015 [90]. In the Tibet Autonomous Region, “exclusive breastfeeding” and “any breastfeeding” rates at discharge were 22% and 75%, respectively, in Lasa in 2016 [91]. Another survey in Sajia showed that “any breastfeeding” rates were 93.50% at six months, declining to 52.03% at 12 months; and no-one was still “exclusively breastfeeding” at six months in 2013 [92].

In Yunnan province, a survey in Kunming showed that “exclusive breastfeeding” and “any breastfeeding” rates at four months were 48.52% and 79.50%, respectively, in 2010 [93]. In Inner Mongolia, a cross-sectional study in Hailaer city in 2014–2015 showed that “exclusive breastfeeding” and “any breastfeeding” rates were 30.16% and 73.81%, respectively, before six months [94]. In the Ningxia Hui Autonomous Region, the “any breastfeeding” rate at four months was 86.15% and the “exclusive breastfeeding” rate at four months was 62.44% in 2009 [95]. In the Guangxi Zhuang Autonomous Region, the “exclusive breastfeeding” rate at 1.5 months was 64.18% in Nanning from 2010 to 2011 [96], and the rate at the same months was 73.33% in Liuzhou in 2014 [97].

In summary, “exclusive breastfeeding” rates were lower in the minority areas. For example, “exclusive breastfeeding” rate at four months was 63.03% in Tuoli, 62.44% in Ningxia and 48.52% in Kunming [90,93,95], and the “exclusive breastfeeding” rate at six month was 30.16% in Hailaer and 0% at Sajia [92,94]. “Any breastfeeding” rates were in a larger range. For example, the “any breastfeeding” rate at four months was 86.15% in Ningxia and 79.50% in Kunming [93,95], and the “any breastfeeding” rate at six months was 93.50% at Sajia and 73.81% in Alaer [92,94].

### 3.6. Changes in Breastfeeding Rates in China from 2007 to 2017

During the latest decade, a series of initiatives and measures have been taken to promote breastfeeding in China, including promoting the Baby Friendly Hospital Initiative, women and child health protection legislation, ten steps to successful breastfeeding, society support programs, breastfeeding education programs, continuous nursing and complementary food control [98,99]. The Baby Friendly Hospital Initiative is used to promote breastfeeding in China [6]. In 1992, only 21 state-owned hospitals in China had received baby friendly hospital initiative certification. By 2015, there were 7036 certified baby-friendly hospitals in China, and 66% of births took place in baby-friendly hospitals [100].

The fourth national health service survey in 2008 showed that the “exclusive breastfeeding” rate at six months was 27.6% [101]; the rate increased to 58.5% in the fifth national health service survey in 2013 [102]. In Shanghai, after promulgation of “the Outline of Development of Chinese Women (2011–2020)”, the “exclusive breastfeeding” rate of 3585 newborn infants in Jiangwan Community of Hongkou District in 2012–2016 was significantly higher than that of 3367 newborn infants in the same area in 2007–2011 (49.6% vs. 26.9%, *P* < 0.05) [98]. A quasi-experimental study (*n* = 16,867) throughout 14 provinces in Eastern, Central and Western China showed that “exclusive breastfeeding” rates at six months increased significantly from 42.96% in 2012 to 48.84% in 2015 after carrying out standard health management for children in these areas [103]. On the other hand, a retrospective study (*n* = 8673) in poverty-stricken areas from 13 provinces of China in 2007–2009 showed that “any breastfeeding” rates at six months decreased from 93.8% in 2007 and 93.5% in 2008 to 91.7% in 2009; and “full breastfeeding” rates at six months also declined slightly from 52.3% in 2007 to 43.4% in 2008 and 48.2% in 2009 [104].

Compared with the results in our previous literature review [6], the proportion of studies measuring breastfeeding rates, including “exclusive breastfeeding” and “any breastfeeding” rates, at six months has increased. This may be associated with the changes in breastfeeding targets set in the National Program of Action for Child Development in China. In the 1990s, the target was breastfeeding of 80% by 2000 and promoting “exclusive breastfeeding” to four or six months [6]. The target from 2001 to 2010 was a breastfeeding rate of 85% and timely introduction of complementary food [6]. The recent target from 2011 to 2020 is an “exclusive breastfeeding” rate of 50% and over at the sixth month of life [5]. The new target is more specific than the previous targets [5].

The main problem found in this review of breastfeeding in China is the “exclusive breastfeeding” rates, which remain low (Figure 3). This is consistent with reports from other countries, with only an average of 38% of infants aged 0 to 6 months being exclusively breastfed globally [105]. The World Health Organization International Code of Marketing of Breastmilk Substitutes (the Code) is critical to protecting exclusive breastfeeding [105]. However, more effort needs to be made to ensure that the Code is followed everywhere in China. A study among 291 mothers with babies under 6 months old from six cities in China found that 40.2% of the mothers reported receiving free formula samples, violations of the Code, with 76.1% received the free samples in or near a hospital [106].

### 3.7. Reasons for Discontinuing Breastfeeding or Exclusive Breastfeeding before Six Months in China

Table 4 shows the reasons for discontinuing breastfeeding or introducing water, formula or other infant food before six months in nine cities (Shanghai [69,80,107], Hangzhou [80], Yuncheng [108], Shenzhen [71], Xining [50,51], Yongkang [29], Xi’an [50], Panzhihua [109] and Kunming [93]). in China. From three studies (in Xi’an and Xining, Shenzhen, and Kunming), the reasons for discontinuing “exclusive breastfeeding” were perceived breastmilk insufficiency (the first reason), mother’s returning to work (the second reason), maternal and child illness (the third reason for Xi’an and Xining and Kunming) and concern about nutrition or available formula milk (the third reason for Shenzhen). From the seven studies in six cities (two studies in Shanghai), the reasons for discontinuing “any breastfeeding” were “perceived breastmilk insufficiency” (the most common reason for five cities except Shanghai) and “mother returning to work” (the first reason for Shanghai, the second reason for other four cities except Panzhihua). See Table 4 for more details. A survey in mothers of 0–3-year-old children in Guangzhou showed that 39.8% of mothers thought their breast milk supply was insufficient, 17.2% of them felt that breastfeeding should be stopped when babies grew to a certain month of age (the cut-off month was not mentioned) [35]. In a cross-sectional study in mothers with 0–2 years old children in Sajia, a county of the Tibet Autonomous Region, 19.6% mothers thought that their babies should be weaned off breastfeeding [92]. In Shandong province, taking the night shift (accounted for 54.19%) was the main reason for weaning breastfeeding among mothers who were nurses [73].

The common reasons given for ceasing breastfeeding or “exclusive breastfeeding” before six months were: perceived breast milk insufficiency, mother returning to work and maternal or child illness. The three reasons were similar to those reported in a previous review [6]. Moreover, concern about nutrition or available formula milk was another common reason for discontinuing breastfeeding in recent years.

There are several limitations that need to be considered when interpreting the results of this literature review. We have endeavoured to find all relevant studies. However, there are only two extensive electronic data bases in the Chinese language. There may be some small regional journals that are not included in these databases. We have relied on the databases, our own knowledge of this field and consultations with colleagues. The study does not include several special areas of China, such as, Macao, Hong Kong and Taiwan. Any studies published in languages other than English or Chinese have not been included in this review. While the National Government and provincial centres for disease controls provide guidance on the measurement of breastfeeding and other public health parameters, there may be some variation in definitions used and methodology across the country. The comparability of studies across time periods in a vast country is a limitation that has to be considered in interpreting the data. The authors have presented the data as recorded in the published literature.

## 4. Conclusions

The mean duration of “any breastfeeding” in China appears to have increased by up to two months in the last decade. The proportion of mothers breastfeeding at four months has increased from 78% to 83%. The second baby has a longer breastfeeding duration than the first. While breastfeeding statistics have improved, the exclusive breastfeeding rate is below the national goal. Breastfeeding education needs to emphasize the benefits of “exclusive breastfeeding” for six months and correct traditional perceptions. The findings of this review show the need to explore further factors that are associated with the early introduction of complementary food to babies and to provide more education on exclusive breastfeeding.

## Figures and Tables

**Figure 1 ijerph-17-08234-f001:**
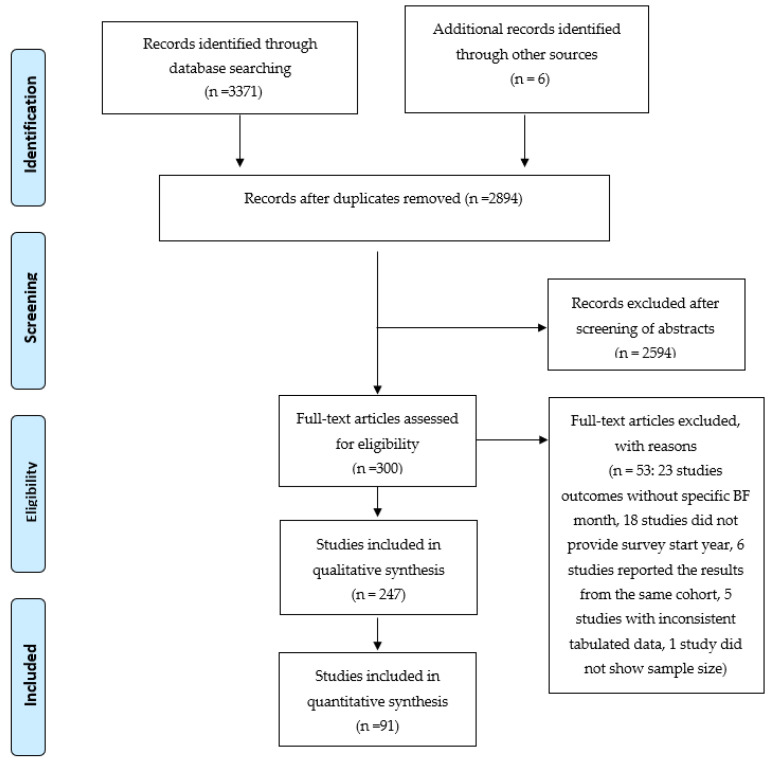
Flow diagram of systematic review following PRISMA protocol.

**Figure 2 ijerph-17-08234-f002:**
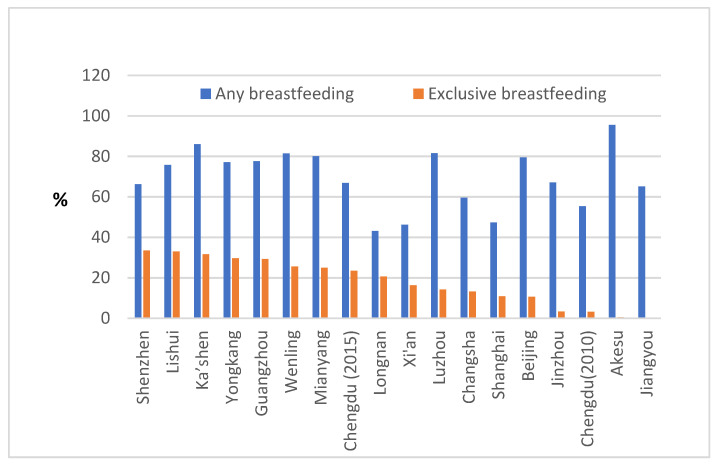
Breastfeeding rates (%) at six months from cohort studies (2005–2016), China.

**Figure 3 ijerph-17-08234-f003:**
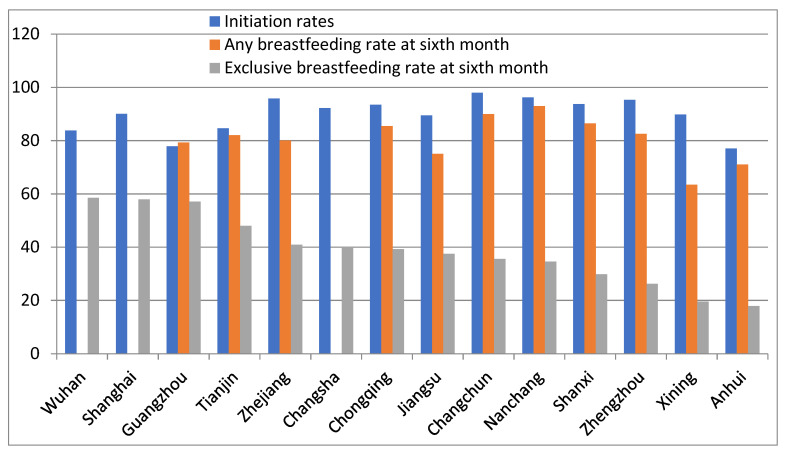
Breastfeeding in China from other studies, 2007–2018.

**Figure 4 ijerph-17-08234-f004:**
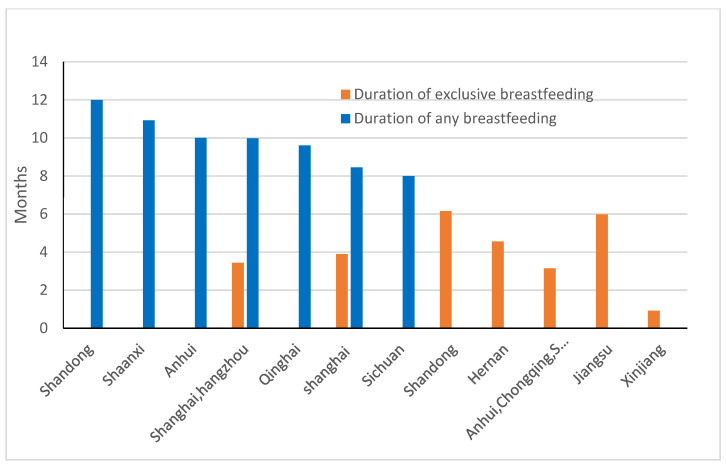
Reported length of breastfeeding in China, 2007–2017.

**Figure 5 ijerph-17-08234-f005:**
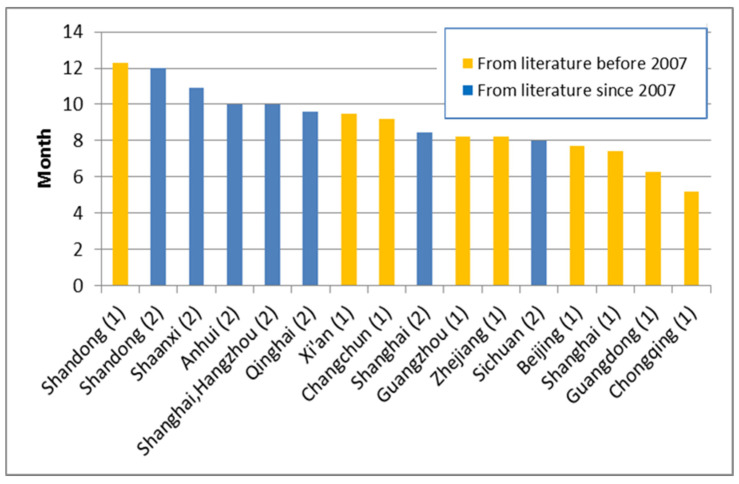
Comparison of length of breastfeeding in China before and after 2007.

**Table 1 ijerph-17-08234-t001:** Breastfeeding rates (%) in the first six months of life from cohort studies, China, 2007–2017.

Study Details	Baby’s Age (month ^a^)	Any Breastfeeding	95% CI		Exclusive Breastfeeding	95% CI	
Beijing [10] *n* = 360 Survey year: 2007–2009	1	95.00	92.75	97.25	64.17	59.22	69.12
	4	89.72	86.58	92.86	40.83	35.75	45.91
	6	79.44	75.27	83.61	10.56	7.39	13.73
Shanghai [11] *n* = 296 Survey year: 2014	0	82.09	77.72	86.46	28.72	23.57	33.87
	1	77.03	72.24	81.82	27.70	22.60	32.80
	3	66.22	60.83	71.61	24.32	19.43	29.21
	6	47.30	41.61	52.99	10.81	7.27	14.35
Chengdu [12] *n* = 438 Survey year: 2015–2016	0	87.90	84.85	90.95	60.96	56.39	65.53
	1	86.07	82.83	89.31	47.95	43.27	52.63
	4	79.45	75.67	83.23	37.90	33.36	42.44
	6	66.89	62.48	71.30	23.52	19.55	27.49
Chengdu [13] *n* = 760 Survey year: 2010–2012	1	88.00	85.69	90.31	60.50 ^b^	57.02	63.98
	3	73.40	70.26	76.54	52.90 ^b^	49.35	56.45
	6	55.40	51.87	58.93	3.20 ^b^	1.95	4.45
Yinchuan [14] *n* = 747 Survey year: 2014–2015	1	94.78	93.18	96.38	54.89	51.32	58.46
	2	92.90	91.06	94.74	51.14	47.56	54.72
	3	83.27	80.59	85.95	35.48	32.05	38.91
	4	80.59	77.75	83.43	27.31	24.11	30.51
Changsha [15] *n =* 228 Survey year: 2012	0	92.11	88.61	95.61	71.05	65.16	76.94
	2	88.60	84.47	92.73	63.16	56.90	69.42
	4	77.60	72.22	83.04	46.05	39.58	52.52
	6	59.60	53.28	66.02	13.16	8.77	17.55
Xi’an [16] *n =* 3580 Survey year: 2013	1	94.80	94.07	95.53	76.51	75.12	77.90
	3	75.20	73.79	76.61	47.09	45.45	48.73
	6	46.20	44.57	47.83	16.31	15.10	17.52
Guangzhou [17] *n =* 416 Survey year: 2005–2006	1	97.60	96.13	99.07	72.12	67.81	76.43
	3	97.12	95.51	98.73	71.39	67.05	75.73
	6	77.64	73.64	81.64	29.33	24.95	33.71
Ka’shen [18] *n =* 300 Survey year: 2015	1	88.67	85.08	92.26	51.33	45.67	56.99
	3	86.67	82.82	90.52	38.00	32.51	43.49
	6	86.00	82.07	89.93	31.67	26.41	36.93
Jinzhou [19] *n =* 972 Survey year: 2014	2	83.13	80.78	85.48	49.07	45.93	52.21
	3	80.86	78.39	83.33	49.07	45.93	52.21
	4	77.06	74.42	79.70	30.56	27.66	33.46
	6	67.08	64.13	70.03	3.29	2.17	4.41
Shenzhen [20] *n =* 325 Survey year: 2013	1	90.15	86.91	93.39	87.38	83.77	90.99
	4	82.71	78.39	87.03	59.66	54.06	65.26
	6	66.20	60.70	71.70	33.45	27.96	38.94
Akesu [21] *n =* 400 Survey year: 2011–2012	0	99.50	98.81	100.19	96.00	94.08	97.92
	1	99.00	98.02	99.98	79.00	75.01	82.99
	3	98.25	96.96	99.54	62.50	57.76	67.24
	6	95.50	93.47	97.53	0.50	0.14	1.80
Luzhou [22] *n =* 486 Survey year: 2012	1	92.59	90.26	94.92	45.06	40.64	49.48
	3	89.71	87.01	92.41	42.80	38.40	47.20
	6	81.48	78.03	84.93	14.2	11.10	17.30
Jiangyou [23] *n* = 695 Survey year: 2010–2011	0	95.10	93.40	96.70	-	-	-
	1	92.70	90.70	94.70	-	-	-
	3	85.10	82.30	87.80	-	-	-
	6	65.10	61.20	69.00	-	-	-
Ma’anshan [24] *n =* 343 Survey year: 2009	2	88.92	85.60	92.24	57.43	52.20	62.66
	3	84.84	81.04	88.64	55.98	50.73	61.23
	4	81.92	77.85	85.99	51.60	46.31	56.89
Mianyang [25] *n =* 1532 Survey year: 2008	1	95.04	93.95	96.13	91.06	89.63	92.49
	4	87.79	86.15	89.43	45.82	43.32	48.32
	6	80.03	78.03	82.03	25.00	22.83	27.17
Ningbo [26] *n =* 318 Survey year: 2008	2	82.39	78.20	86.58	36.00	30.58	41.12
	3	66.67	61.49	71.85	36.00	30.58	41.12
	4	50.63	45.13	56.13	23.60	18.91	28.25
	5	47.80	42.31	53.29	10.00	13.8	13.30
Wenling [27] *n =* 500 Survey year: 2014	0	91.80	89.40	94.20	80.20	76.71	83.69
	1	91.00	88.49	93.51	70.20	66.19	74.21
	4	86.00	82.96	89.04	52.60	48.22	56.98
	6	81.40	77.99	84.81	25.60	21.77	29.43
Lishui [28] *n =* 208 Survey year: 2014	1	89.90	85.80	94.00	34.62	28.15	41.09
	3	84.82	79.73	89.91	34.62	28.15	41.09
	6	75.68	69.50	81.86	32.97	26.20	39.74
Yongkang [29] *n =* 667 Survey year: 2013	1	92.50	90.50	94.50	78.11	74.97	81.25
	3	89.06	86.69	91.43	58.92	55.19	62.65
	6	77.06	73.87	80.25	29.69	26.22	33.16
Longnan [30] *n* = 480 Survey year: 2009	1	93.13	90.87	95.39	78.75	75.09	82.41
	3	68.13	63.96	72.30	42.29	37.87	46.71
	6	43.13	38.70	47.56	20.63	17.01	24.25

^a^: 0 month refers to the time at discharge from hospital for birth, generally at one or two weeks after birth. ^b^: Full breastfeeding. Beijing: the capital city of China. Shanghai: large city, Eastern China. Chengdu: the capital city of Sichuan province, Southwest China. Yinchuan: the capital city of Ningxia Autonomous Region, Northwest China. Changsha: the capital city of Hunan province, South China. Xi’an: the capital city of Shaanxi province, Northwest China. Guangzhou: the capital city of Guangdong province, Southeast China. Ka’shen: a medium-sized city in Xinjiang Uygur Autonomous Region, West China. Jinzhou: a medium-sized city in Liaoning province, Northeast China. Shenzhen: a medium-sized city in Guangdong province, Southeast China. Akesu: a medium-sized city in Xinjiang Uygur Autonomous Region, West China. Luzhou: a medium-sized city in Sichuan province, Southwest China. Jiangyou: a medium-sized city in Sichuan province, Southwest China. Ma’anshan: a medium-sized city in Anhui province, Southeast China. Mianyang: a medium-sized city in Sichuan province, Southwest China. Ningbo: a medium-sized city in Zhejiang province, East China. Wenling: a small city in Zhejiang province, East China. Lishui: a small city in Zhejiang province, East China. Yongkang: a small city in Zhejiang province, East China. Longnan: a small city in Gansu province, Northwest China.

**Table 2 ijerph-17-08234-t002:** “Any breastfeeding” rates at 12 months after birth from cohort studies, China, 2010–2014.

Province or Big City	Survey Commencement	Women (*n*)	Any BF Rate (%)	95% CI	
Shaanxi [33]	2014	1350	73.26	70.90	75.62
Anhui [34]	2012	1332	27.40	25.00	29.80
Shanghai [11]	2014	296	14.19	10.21	18.17
Sichuan [23]	2010	695	12.90	9.90	15.80
Guangzhou [35]	2013	383	10.97	7.84	14.10

BF, breastfeeding.

**Table 3 ijerph-17-08234-t003:** Breastfeeding rates (%) in China 2007–2018.

Study Site	Initiation Rate	Any Breastfeeding Rate at Six Months	Exclusive Breastfeeding Rate at Six Months	Study Details
Changchun * [52,53]	98.00	89.97	35.53	a. Cross-sectional study 2013, *n* = 349; b. Cohort study 2007–2010, *n* = 1600
Nanchang * [54]	96.21 ^a^	92.93	34.53	Cohort study 2011–2013, *n* = 976
Zhejiang ^p^ [55,56]	95.80	80.00	40.89	a. Retrospective study 2015–2017, *n* = 429; b. Cross-sectional study 2013, *n* = 675
Zhengzhou * [43,44]	95.26	82.50	26.25	a. Cross-sectional study 2011–2012, *n* = 612; b. 2013–2014, *n* = 800
Shanxi ^p^ [57,58,59]	93.75	86.45 ^b^	29.78	a. Cross-sectional studies 2016, *n* = 1193 b. 2014, *n* = 487 c. 2009, *n* = 240
Chongqing ** [41,42]	93.49	85.47	39.25	a. Cohort study 2015–2016, *n* = 215; b. 2016–2018, *n* = 57,382
Changsha * [49]	92.20	-	40.00	a. Cross-sectional study 2014, *n* = 1014
Shanghai ** [39,40]	90.00	-	57.91	a. Cross-sectional study 2016, *n* = 200; b. Cohort study 2014, *n* = 815
Xining * [50,51]	89.80	63.41	19.51	a. Cross-sectional study 2012–2015, *n* = 1148; b. Cohort study 2013, *n* = 287
Jiangsu ^p^ [62,63]	89.46	75.00	37.50	a. Cross-sectional study 2014–2015, *n* = 320; b. Cohort study 2010–2013, *n* = 759
Tianjin ** [37,38]	84.60 ^a^	82.00	48.00	a. Cross-sectional study 2015, *n* = 818 b. Cohort study 2011–2012, *n* = 200
Wuhan * [45,46]	83.75	-	58.50	a. Cross-sectional studies 2013–2015, *n* = 2000; b. 2016–2017, *n* = 494
Guangzhou * [47,48]	77.90	79.24	57.12	a. Cross-sectional studies 2011–2014 *n* = 1180; b. 2013–2014, *n* = 289
Anhui ^p^ [34,60,61]	77.02	71.02	17.87 ^c^	a. Cross-sectional studies 2008, *n* = 1736; b. Cohort study 2012–2013, *n* = 1332; c. 2008–2010, *n* = 2747

^a^: At 1st month. ^b^: At 4–6 months. ^c^: Full breastfeeding; ^p^: Province; * Capital city of province. ** Large city (municipality directly under the Central Government). In the column “Study Details”, where there is more than one study from the same city, they are labelled a, b and c.

**Table 4 ijerph-17-08234-t004:** Reasons for discontinuing breastfeeding in the first six months after birth.

Research Site	Insufficient Breast Milk (%)	Return to Work (%)	Maternal or Child Illness (%)	Mother Dislikes Breastfeeding (%)	Concerns on Nutrition of Breastmilk or Available Formula Milk (%)	Other Reason(s) (%)
Xi’an and Xining * (*n* = 240, 2013, [50])	76.25	8.33	6.25	-	3.75	5.42
Shenzhen * (*n* = 388, 2015, [71])	42.01	17.01	-	-	10.31	8.25
Kunming *(*n* = 216, 2010, [93])	38.0	23.9	22.6	5.8	-	7.9
Panzhihua ** (*n* = 293, 2014, [109])	62.4 (R)		18.3 (R)	32.1 (R)	-	27.1 (R)
Yuncheng ** (*n* = 1200, 2016, [108])	56.89 (U) 40.15 (R)	14.08 (U) 8.88 (R)	3.81 (U) 4.25 (R)	2.35 (U) 4.05 (R)	10.41 (U) 22.20 (R)	12.47 (U) 21.27 (R)
Xining ** (*n* = 1148, 2012–2015, [50])	44.6	12.5	9.4	1.7	13.2	18.5
Hangzhou, Shanghai **(*n* = 1046, 2017, [80])	38.5	23.1	6.2	-	22.9	32.2
Yongkang ** (*n* = 274, 2014, [29])	32.85	18.25	8.39	-	-	22.63
Shanghai ** (*n* = 5672, 2017, [69])	26.90	32.75	18.87	-	-	21.48
Shanghai ** (*n* = 272, 2013–2014, [107])	14.70	26.10	-	-	8.82	6.25

* Reasons for discontinuation of “exclusive breastfeeding”. ** Reasons for discontinuation of “any breastfeeding”. U: urban; R: rural.

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
