# Peer review of "Breastfeeding in China: A Review of Changes in the Past Decade"

_ijerph, 2020, doi:10.3390/ijerph17218234_

Round 1

Reviewer 1 Report

In this paper the authors give an extensive review of papers on the rate of breastfeeding in China for the period 2007-2018. The data of this period are compared with a previous survey of the period before 2007. Data are interesting, however I do have some comments.

  1. The present survey is based on cohort and "other" studies. I miss any discussion about the quality of the data in these studies. For a cohort study, were participants aware that the aim of thge study was to investigate the duration of breastfeeding? If yes, might this cause an important bias? What was the aim of other studies? 
  2. How well are studies done before 2007 comparable with the studies after 2007? Breastfeeding is dependent on many factors, like education and social status. The design of the study is also important, as the authors indicate was the follow-up period of the studies before 2007 shorter than in the period > 2007. Are studies comparable regarding regarding important factors like education of the mother, social status etc? How did a shorter follow-up period influence the results?
  3. Rates of breastfeeding are rather variable in a huge country like China. Is it correct to make a conclusion for the whole of China?
  4. On the basis of which data do you conclude that the rate of breastfeeding increased with two month in the second compared to the first period? I can not find the basis for this conclusion.
  5. You did not study the potential role of formula in relation with the duration of breastfeeding, so the statement on the role of formula in the conclusion, both in the abstract and in the paper, should be deleted.

Author Response

We would like to thank the reviewers for the hard work put into reviewing our paper.  We have responded to all of the reviewers’ comments and we believe the paper s much improved as result of their input.

Reviewer 1

In this paper the authors give an extensive review of papers on the rate of breastfeeding in China for the period 2007-2018. The data of this period are compared with a previous survey of the period before 2007. Data are interesting; however I do have some comments.

  1. The present survey is based on cohort and "other" studies. I miss any discussion about the quality of the data in these studies. For a cohort study, were participants aware that the aim of the study was to investigate the duration of breastfeeding? If yes, might this cause an important bias? What was the aim of other studies? 

We assessed the quality of studies using the method from our most recent review based on the STROBE guidelines and the reference has been added to the text  (Zhao 2017, REF 8) The sentence below has been added to method section (page 2, paragraph 3).

“All of the studies were assessed for quality using a method based on the STROBE criteria and used by us in a previous meta-analysis in China (1). This method allocates a maximum score of 18 and papers that met a minimum score of 11 were included”.

‘Whether your baby was breastfed ‘and ‘how long your baby was breastfed ‘are basic questions for breastfeeding studies.  In these studies, they were asked retrospectively.  All mothers who participate in breastfeeding studies would be asked these questions. There is no evidence that mothers who participate in long term breastfeeding studies are more likely to breastfeed their babies for longer than those who did not participated the studies.

We do not consider that that there is any likelihood of bias due to the questions asked.  All long-term studies of breastfeeding have the same aim – simply to measure duration. 

  1. How well are studies done before 2007 comparable with the studies after 2007? Breastfeeding is dependent on many factors, like education and social status. The design of the study is also important, as the authors indicate was the follow-up period of the studies before 2007 shorter than in the period > 2007. Are studies comparable regarding important factors like education of the mother, social status etc? How did a shorter follow-up period influence the results?

Thankyou for the comment.

The quality of studies over long periods of time across a vast nation is a limitation that is included in the limitations paragraph.  We can only report the data as published. We have added the following sentences to the limitations paragraph:

“While the National Government and provincial centres for disease controls provide guidance on the measurement breastfeeding and other public health parameters there may be some variation in definitions used and methodology across the country. The comparability of studies across time periods in a vast country is a limitation that has to be considered in interpreting the data.  The authors have presented the data as recorded in the published literature.”

The follow-up period is unlikely to have influenced the results. In breastfeeding cohort studies, the follow up is usually intended to give proprtions breastfeeding at a specified time, usually 4 and 6 months and less often12 months.  Increasingly cohort studies are measuring median duration. However this was not done consistently in the Chinese studies. The data before and after 2007 are believed to be comparable because in the selected studies, similar definitions and parameters of breastfeeding were used.

Breastfeeding is impacted by a number of factors including education level, social status, rural/urban location and culture.  And the impact of the factors may be different in different studies. This may contribute to the diversity of breastfeeding rates and durations from different studies. On the other hand, this literature review is based on existing studies which aimed to show breastfeeding rates and duration.  In undertaking any review of previous studies, the writers take the methods of the original authors at their face value. In both reviews the quality of the papers has been assessed.  This has now been added to the limitations paragraph.

  1. Rates of breastfeeding are rather variable in a huge country like China. Is it correct to make a conclusion for the whole of China?

We agree that breastfeeding rates vary across such a large nation.  However, in China infant feeding policies (and other public health initiatives) are prescribed by the National Government and Provinces are expected to promoted these goals. This may mean that breastfeeding rates are influenced by known factors mentioned above. We believe that the range of studies in this review are sufficient to give a reasonable representation of trends in breastfeeding rates in China. We have included the following sentence in the abstract

“There appears to have been a modest increase in breastfeeding in China”

“We have included a sentence in the limitations section (see previous paragraph) and added the following to the Conclusion

“The mean duration of breastfeeding in China appears to have increased by up to two months in last decade.”

  1. On the basis of which data do you conclude that the rate of breastfeeding increased with two month in the second compared to the first period? I cannot find the basis for this conclusion.

Thankyou for this comment as it is central to our assessment of breastfeeding progress in China We have checked the data in the original study reports and in addition we have calculated and included the proportion of mothers breastfeeding at 4 months, since this statistic is common to almost all of the reports. This statistic also supports our conclusion that there has probably been a modest increase in breastfeeding in China as reported in the literature. The relevant paragraph

 (page 10, 2nd paragraph) has been rewritten in the following way:

“Figure 4 shows the mean length of breastfeeding from 11 studies in China conducted between 2007and 2017(2-13). The duration of ‘any breastfeeding’ was between eight and 12 months (9 to 11 months in the majority of cities) reported from cross sectional studies. The mean duration was 10.1 months (weighted by provincial population) from 8 cross sectional studies.  the. In the previous study the weighted mean duration of breastfeeding was 8.0 months from cross sectional studies. The reported mean duration of breastfeeding is an increase of almost two months compared with the previous review., 7 to 9 months in majority cities)(14). For example, ‘any breastfeeding’ duration in Shanghai was 7.4 months in between 1999 and 2002(14), and has increased to 8.45 and 9.98 months (Figure 3 and 4). Breastfeeding duration reported from studies before and after 2007 are shown in Figure 5.

The proportion of mothers breastfeeding at 4 months were reported in both decades in the cohort studies. The weighted (on Provincial population) mean rate of 4 months ‘any breastfeeding’ was 78.2% before 2007 and 83.0% more recently”

  1. You did not study the potential role of formula in relation with the duration of breastfeeding, so the statement on the role of formula in the conclusion, both in the abstract and in the paper, should be deleted.

Accepted. “Formula” has been deleted from abstract and conclusion.

Reviewer 2 Report

This article is generally well written and clear, but there are some grammatical and stylistic adjustments that are needed throughout.

Did any of the articles provide information regarding the use of mothers own pumped milk fed to babies artificially (bottle, spoon, etc)?  If so, is there any indication that pumped milk feeding has increased in recent years?

Page 7:  This information is important and useful, but would be better presented in a table instead of in a narrative format.

It would be of interest to other investigators and clinicians if you could compare the five communities with the highest breastfeeding rates with the five lowest rates for the differences in services provided to support breastfeeding.

Page 12:  The last sentence of the first paragraph is not clear.  Please revise.

The Baby Friendly Hospital Initiative is mentioned, but no information is provided regarding the effect of this implementation on breastfeeding rates in various districts or over time.  This information would be of importance for policy decisions.

Author Response

We would like to thank the reviewers for the hard work put into reviewing our paper.  We have responded to all of the reviewers’ comments and we believe the paper s much improved as result of their input.

This article is generally well written and clear, but there are some grammatical and stylistic adjustments that are needed throughout.

Did any of the articles provide information regarding the use of mothers own pumped milk fed to babies artificially (bottle, spoon, etc)?  If so, is there any indication that pumped milk feeding has increased in recent years?

Thanks for the question.

We did not include this in the objectives of our review.  Outside of neonatal intensive care units in large urban hospitals this is not a common practice in China. Although the information is not included in our review one study did mention pumping milk by mothers of infants in Neonatal Intensive Care Units (15)。

Page 7:  This information is important and useful, but would be better presented in a table instead of in a narrative format.

 Because of inconsistent in measures between the studies it is felt to be more appropriate to include the information in its present format.

It would be of interest to other investigators and clinicians if you could compare the five communities with the highest breastfeeding rates with the five lowest rates for the differences in services provided to support breastfeeding.

 Thanks for the comments! This is a good advice for our next study which focuses on factors associated with breastfeeding in China. Services will be considered.

Page 12:  The last sentence of the first paragraph is not clear.  Please revise.

 Accepted. The sentence has been changed to:

“Other risk factors associated with  breastfeeding duration included mother’s education level (mothers with high education level were more likely to be employed and had shorter breastfeeding duration), grandparents’ support ( babies were breastfed shorter if grandparents helped to take care of them)  and having breastfeeding difficulties(babies were less likely to be breastfed if their mother had breastfeeding difficulties(13)”.

The Baby Friendly Hospital Initiative is mentioned, but no information is provided regarding the effect of this implementation on breastfeeding rates in various districts or over time.  This information would be of importance for policy decisions.

Thanks for the comments! The information has been added to the manuscript:

“The Baby Friendly Hospital Initiatives promote breastfeeding in China significantly(14). In 1992, only 21 state-owned hospitals in China started baby friendly hospital initiative (received the baby-friendly certification).  By 2015, there were 7,036 certified baby-friendly hospitals in China and 66% births took place in baby-friendly hospitals(16)”.

Reviewer 3 Report

This is a comprehensive, well-designed review of breastfeeding rates in China. Such a study is important to determine whether changes in government policies have actually changed breastfeeding rates. It is important for China and for other countries to see that breastfeeding rates have increased. This collation of data provides this important evidence.

Author Response

We would like to thank the reviewers for the hard work put into reviewing our paper.  We have responded to all of the reviewers’ comments and we believe the paper s much improved as result of their input.

Reviewer 3

This is a comprehensive, well-designed review of breastfeeding rates in China. Such a study is important to determine whether changes in government policies have actually changed breastfeeding rates. It is important for China and for other countries to see that breastfeeding rates have increased. This collation of data provides this important evidence.

Thanks for the comments!

Round 2

Reviewer 1 Report

The authors have responded  to all my concerns. I still have three remarks.

Remarks. 1. In the conclusion of the abstract there is still a sentence on formula feeding, that needs to be deleted as it is not investigated in this study.

2. Figure 5 is confusing to me, either delete it or give more clear explanation. When looking at the figure, one must draw the conclusion that there is no change in duration of breastfeeding

3. I feel that in the conclusion at the end op the paper it must be made clear that the longer duration of breastfeeding is related to any breastfeeding. The word any needs to be added. 

Author Response

Manuscript ID: ijerph-961127

Thankyou to the reviewer for reading our manuscript again 

Response to Reviewer One

  1. In the conclusion of the abstract there is still a sentence on formula feeding, that needs to be deleted as it is not investigated in this study”

This was already deleted in the first round.  It is not in the version sent to us for review

  1. Figure 5 is confusing to me, either delete it or give more clear explanation. When looking at the figure, one must draw the conclusion that there is no change in duration of breastfeeding

“Breastfeeding duration reported from studies before and after 2007 are shown in Figure 5”

This has been changed to “The mean duration of ‘any’ breastfeeding reported from studies before and after 2007 are shown in Figure 5. This shows that the majority of studies reporting longer duration of breastfeeding have been published since 2007”.

  1. “I feel that in the conclusion at the end op the paper it must be made clear that the longer duration of breastfeeding is related to any breastfeeding. The word any needs to be added.”

Thankyou.  We have added ‘any’ to the conclusion.  We have also added ‘any’ to the abstract for consistency.  

Revised manuscript is attached